# Magnetic Bioreactor for Magneto-, Mechano- and Electroactive Tissue Engineering Strategies

**DOI:** 10.3390/s20123340

**Published:** 2020-06-12

**Authors:** Nelson Castro, Margarida M. Fernandes, Clarisse Ribeiro, Vítor Correia, Rikardo Minguez, Senentxu Lanceros-Méndez

**Affiliations:** 1BCMaterials, Basque Centre for Materials, Applications and Nanostructures, UPV/EHU Science Park, E-48940 Leioa, Spain; senentxu.lanceros@bcmaterials.net; 2Centre of Physics, University of Minho, Campus de Gualtar, 4710-057 Braga, Portugal; margaridafernandes@fisica.uminho.pt (M.M.F.); cribeiro@fisica.uminho.pt (C.R.); 3Centre of Biological Engineering, University of Minho, Campus de Gualtar, 4710-057 Braga, Portugal; 4Algoritmi Research Centre, University of Minho, Campus de Azurém, 4800-058 Guimarães, Portugal; vcorreia@dei.uminho.pt; 5Department of Graphic Design and Engineering Projects, University of the Basque Country, E-48013 Bilbao, Spain; 6IKERBASQUE, Basque Foundation for Science, E-48013 Bilbao, Spain

**Keywords:** magnetic bioreactor, magnetoactive scaffolds, tissue engineering, magnetic actuator, magnetoelectric stimulation

## Abstract

Biomimetic bioreactor systems are increasingly being developed for tissue engineering applications, due to their ability to recreate the native cell/tissue microenvironment. Regarding bone-related diseases and considering the piezoelectric nature of bone, piezoelectric scaffolds electromechanically stimulated by a bioreactor, providing the stimuli to the cells, allows a biomimetic approach and thus, mimicking the required microenvironment for effective growth and differentiation of bone cells. In this work, a bioreactor has been designed and built allowing to magnetically stimulate magnetoelectric scaffolds and therefore provide mechanical and electrical stimuli to the cells through magnetomechanical or magnetoelectrical effects, depending on the piezoelectric nature of the scaffold. While mechanical bioreactors need direct application of the stimuli on the scaffolds, the herein proposed magnetic bioreactors allow for a remote stimulation without direct contact with the material. Thus, the stimuli application (23 mT at a frequency of 0.3 Hz) to cells seeded on the magnetoelectric, leads to an increase in cell viability of almost 30% with respect to cell culture under static conditions. This could be valuable to mimic what occurs in the human body and for application in immobilized patients. Thus, special emphasis has been placed on the control, design and modeling parameters governing the bioreactor as well as its functional mechanism.

## 1. Introduction

Fundamental biological studies and therapeutic applications rely on tissue engineering (TE) techniques, which aim to mimic the physicochemical and bioactive characteristics of natural cellular matrices [1,2], in order to achieve the replacement and/or regeneration of damaged tissues or organs [3,4]. When building a new tissue culture, three tools are mainly used: cells, scaffolds and stimuli. The cells are the building blocks for tissue culture as they contain the pre-programmed information that allows tissue regeneration. Cells are thus placed in a scaffold, which acts as the cell culture support, where the necessary environment is present, mainly in terms of biochemical stimuli, through the inclusion of growth factors and biophysical stimuli, by using a bioreactor [5]. Bioreactors allow us to introduce different chemical or physical stimuli on tissue culture, depending on both bioreactor technology and on how scaffold structures respond to those stimuli, in order to create a synergistic environment, thus stimulating cell response. Biomaterials used as scaffolds can be tailored, allowing them to be passively tolerated by the organism or actively providing the most appropriate and specific cell responses [6,7].

In the context of physically active TE by using bioreactors, different approaches have been implemented, including mechanoactive and electroactive scaffolds, among others [1,8]. In particular, electroactive materials are gaining increased attention due to the possibility of regulating different cell functions by providing electrical signals to the tissue culture [9,10,11]. Examples of such materials are piezoelectric scaffolds, which provide the necessary stimuli for the effective regeneration of bone [7,9,12], neural tissue [13,14], muscle [15,16,17], among others. The underlying mechanism relies on the possibility for mechanoelectrical transduction from materials to the cells [12], but lack of appropriate bioreactors able to stimulate those materials and take full advantages of their smart and multifunctional nature. In this work, a bioreactor is presented able to apply magnetic, mechanical and electrical stimuli to the cells in culture, based on the application of a magnetic field to a magnetically responsive scaffold containing magnetostrictive nano/microparticles embedded in a specific matrix [18,19] which can be electrically responsive (e.g., piezoelectric [20]) or not. These stimuli can be important for different types of tissue and, in particular, for bone TE due to the piezoelectric characteristics of bone [21], allowing to address novel TE strategies [22]. In particular, the performance of bone tissue along with its development and functional characteristics is strongly influenced by the voltage variation generated by mechanical stress to which the bone is subjected and, therefore, piezoelectric stimuli must be considered for proper regeneration strategies [23].

In fact, the electric sensitivity of osteoblasts has been regarded as an important tool for enhancing the ossification and healing through electric stimulation, as proven by piezoelectric scaffolds stimulated by a mechanical bioreactor, thus providing a proper electroactive environment to the cells [24,25]. In a similar approach, conductive composites have been proven to deliver exogenous electric currents to cells and increase their function [26], while evidence of electric stimulation influence on the ossification has been indeed observed [27].

In a novel approach, magnetoelectric (ME) scaffolds were used in order to provide support for cell seeding and proliferation while taking advantage of the scaffolds material, which enables electromechanical stimuli to the cells, generated by a magnetic field [19,28,29]. These composites are composed of magnetostrictive and piezoelectric layers working synergistically to produce the ME effect [30]. The ME effect can be described as a transduction from a magnetic field to an electrical field once the vibration of the magnetostrictive phase generated by an alternated magnetic field results in an electrical charge variation at the piezoelectric phase terminals at room temperature [31,32]. Further, if the support is not piezoelectric, just a mechanical vibration will be induced, providing such stimuli to the neighboring cells [33].

The flexibility, versatility and biocompatibility of these materials [8,34] can take advantage of in-vitro dynamic cultures through the support of a remote magnetic field [33]. Thus, materials with ME properties are therefore regarded as breakthrough platforms for TE applications that allow for remote generation of these physical stimuli, resulting in a controled influence on the surrounding tissue [9]. This effect has already been proven to induce a magnetomechanical [20] or a local magnetomechanical and magnetoelectrical effect [19], on the cells thus triggering improved cell proliferation and differentiation effect.

It is important to notice that the application of the magnetic field by itself, without further magnetomechanical or magnetoelectrical transduction, is also interesting for biomedical applications. This enables stimulation of cellular functions and cell manipulation to create cellular clusters, enabling more complex tissue structures than conventional strategies based on static scaffolds [35]. As a recent example, the use of superparamagnetic iron oxide nanoparticles proved to be a promising bioactive additive for scaffold fabrication [36], the scaffold enhancing the performance of human dental pulp stem cells yielding a higher count of phosphatase activity, higher osteogenic marker gene expression and improved cell-synthesized bone minerals. Other methods include the marking of C2C12 cells with magnetite cationic liposomes, mixed in a collagen solution, and seeded in a cell culture space of a hollow-fiber bioreactor [37]. The results demonstrated that high cell-density and viable tissue constructs containing myotubes were successfully obtained. Magnetic stimuli through permanent magnetic displacement were also proposed [38]. Rotation of permanent magnets was also employed in order to induce cellular growth proving that the variation of the magnetic field between 7 and 10 Hz increased the growth of neurite on chromaffin cells [39]. These devices can thus give an important contribution to the field, in order to overcome the issues related to the traditional cell culture conditions, improving the cellular distribution and accelerating cellular growth [40].

Besides biocompatibility and sterility, the design of mechanical bioreactors requires accurate control of the applied stimulus in order to get accurate data and to replicate results within the same parameters. In order to comply with these requirements and build a user-friendly device, which enables the user to apply controlled stimuli while providing control over temperature, culture stimuli active and resting time schedule, as well as total time, a modular magnetic bioreactor with an interchangeable magnets table was designed and developed. The interchangeable magnets table module has been designed to be used with 24-multiwell standard plates that can be easily operated and calibrated by the user. Magnetic stimulation has been previously reported [19,20,41], however, the present work reports on the development and validation of a novel bioreactor for magnetic stimulation of cells and/or scaffolds, comprehending cells-scaffold-stimuli relation, electromechanical study for actuation, control user interface and cell culture validation.

In terms of practical applications, the herein developed bioreactor can be a valuable tool for novel and more efficient tissue engineering strategies, including (i) to perform cell culture assays in vitro to validate the use of magneto-active materials for tissue engineering, thus avoiding extensive animal testing; (ii) to grow in vitro cellular tissue to be further implanted in a patient, after detaching the tissue from the surface of a magneto-active material and (iii) to be used in vitro to grow cellular tissue that is further implanted in a patient with the magneto-active material, which would allow for a remote stimulation of the material in the body, an important tool for immobilized patients.

## 2. Bioreactor Design

A bioreactor is an important tool for TE purposes since it allows us to mimic essential elements of the tissue environment and thus evaluate the influence of the stimuli on cell proliferation and differentiation. In order to study the influence of the electromechanical stimulation in bone tissue cells remotely [19,20], magnetoelectric scaffolds have been developed to provide the required stimuli [19], where an alternated magnetic field was applied to supply the necessary magnetic stimulation to the scaffolds. In this way, the herein developed device was designed to fulfill these characteristics and to meet experimental requirements to which the bioreactor will be subjected, which includes an incubation chamber with controlled temperature and humidity (37 °C and 95%, respectively). Thus, increasing the scaffolds temperature while applying the magnetic field is a critical design parameter, excluding the use of electromagnets, which requires high current flow, consequently resulting in radial heat from windings [42]. On the other hand, the displacement of permanent magnets as magnetic actuators on magnetoelectric scaffolds were used instead, to avoid a bulkier system and further heat. A schematic representation of the designed system is presented in Figure 1. The actuation system is composed of a permanent magnetic table that is displaced at a controlled frequency until certain limits, in order to get the required alternated magnetic field at the culture plate. For that, a mechanical structure comprising a motor in a ball screw assembly was installed to obtain an electromechanical actuation system. For mechanical protection, limit switches and precision sensors were applied to obtain electronic control of table position through a linear sensor and magnetic encoder for speed, as main operation components.

Furthermore, the system required a stable power supply in order to handle all digital and power electronic components, as well as remote control through wireless communication such as Bluetooth^®®^ and respective user-machine interface through buttons and light and/or display feedback. To facilitate the use of this bioreactor in TE, the design included the application of a commercially available culture plate on the top of the system, where the ME scaffolds are easily placed and tested for cell culture as represented in Figure 1. The starting position of the magnets is user-defined as well as its displacement and motion frequency. It is important to note that the displacement will influence the magnetic variation over time in a specific place of the culture wells. The magnetic field peak values are directly dependent on magnets grade and distance to the scaffold, which can be calibrated mechanically by adding extra layers below the magnet table. The required magnetic field for the stimulation of the ME samples must reach values within a range of 20 to 50 mT [20]. The selected magnets are nickel-plated neodymium disks S−15–03-N52N from Supermagnete. The distance between the culture bottom and the N52 grade neodymium permanent magnets influences the magnetic field intensity, as analyzed through simulation with ANSYS^®®^ Software. The magnetic field generated by a permanent magnet is easily stronger at a given z distance from the source, by comparison with fair current amplitude within a reasonable size coil for this system. It can be calculated for a cylinder type of permanent magnet, using Equation (1).
(1)$$B=\frac{B_{r}}{2}\left(\frac{D+z}{\sqrt{R^{2}+{\left(D+z\right)}^{2}}}-\frac{z}{\sqrt{R^{2}+z^{2}}}\right).$$

This way, magnetic flux density can be calculated at a certain distance, where *Br* is the remnant field, independent of the magnet’s geometry, *z* the distance from a pole face on the symmetrical axis, *D* the thickness (or height) of the cylinder and R the semi-diameter (radius) of the cylinder [43]. Figure 2 displays the more appropriate distance according to each culture wells plates setup, which is different according to the number of magnets and radius used, resulting in a distance of 10 mm, where a field intensity of 30 mT was achieved at culture plate bottom (Figure 2a). In Figure 2b, it is possible to observe the side views with a larger range of the magnetic field variation, due to the substantial higher field at the N52 permanent magnets core. This fact enables higher magnetic fields at the culture bottom by reducing distance through magnetic table mechanical elevation in the same mechanism represented in Figure 2c. Furthermore, tailoring/exchanging the permanent magnets table with different magnetic grades, sizes and geometries, enables the mechanics to fit with the number of wells and geometries that a culture plate may present.

As schematized in Figure 1, the DC motor is controlled by a custom electrical system built and designed for the purpose of this application (control system). For this role, the system takes advantage of the sensors (limit switches, encoder and linear) in order to perform a close loop control of the magnets table positioning and displacement frequency. Furthermore, the system handles a user firmware interface that allows us to select the culture cycles for active and resting times. Regarding the firmware tasks, they are divided into two main control units: one unit controls the user inputs while the other controls the magnets positioning, sensors and display feedback. The local user interface is composed of an ILI9341 LCD providing 240 × 320 resolution with 262 k color and a side capacitive touch wheel with one button and a sliding circular panel for selection and option confirmation. The setup allows the user to locally stop, start or change the culture control parameters. A remote interface was also installed by Bluetooth, which allows us to monitor the culture status and the sensor reads from outside the incubator, using a mobile terminal. Since cell culture experiments require aseptic environments, requiring a sterilization process before each cell culture experiment, the creation of a waterproof enclosure was necessary. Nevertheless, such enclosure should withstand the temperature without overheating the bioreactor thus damaging the cell culture. The three-dimensional project of the device complying with such requirements is presented in Figure 3.

The magnetostriction of the scaffolds is obtained as a consequence of the magnetic field application, achieved by the movement of the magnets along the horizontal axis of the magnets table, below the cell culture plate. The selected construction material was nylon in order to avoid magnetic field interference. The system was designed for 24 and 48 well culture plates, although several types of commercial plates can be used with a different number of culture wells. Further, it was designed to be modular, thus the permanent magnets table can be replaced with a higher or lower number of magnets to fit the culture plate and apply an even magnetic field to all scaffolds (Figure 3). The result of the assembled system is simple, compact and a sealed design (Figure 3a). It is worth noting that the herein used permanent magnets were selected according to the magnetic field level required for inducing an electroactive environment on the ME scaffolds. Thus, when subjected to a magnetic field, a magneto-mechanical and magneto-electric stimulation is induced on the scaffold due to the incorporated magnetostrictive particles [19,44]. The magnetostrictive particles deform and generate the mechanical stimulus to the piezoelectric polymer within the scaffold which, in turn, develops an electrical charge that passes to the cells. The movement of magnets along the horizontal axis is achieved by using two side supports and a central motor shaft coupled with a DC motor, a component that can be observed in the device sectional detail in Figure 3d. The mechanical setup is implemented with a 25GA370 DC motor with 400 rpm and is controlled through an H-Bridge at 20 kHz pulse width modulation (PWM) pulses. This was selected over a stepper motor due to the possibility of applying lower currents and avoid heating in order to keep the system temperature low to protect the cell culture. The magnet table position is controlled by a linear sensor 9615R5.1KL2.0 (from BEI Sensors, Attleboro, MA, USA), together with an ADC resolution of 12 bits, which results in a linear resolution of 0.01 mm. The bioreactor was designed with an IP68 waterproof rating system to withstand the sterilization process, through waterproof power connector, rubber protection in the joints and capacitive touch avoiding mechanical buttons and leaks. Figure 3 displays a photo of the prototype device after the design and development phase.

### 2.1. Power and Control Circuitry

The main system communication and control between modules is illustrated in the block diagram of Figure 1. Thus, an electrical system was developed according to the mechanical design and user interface operational requirements. Further, the designed circuits were implemented using commercial electronic components. The electrical circuits designed are represented in the schematic of Figure 4. These circuits can be divided between power conversions (A), user interface (B), sensors (C), main control of operation (D), actuation (E) and system wireless communications (F). Power conversion is required in order to comply with three different voltage levels, whereas the motor operates with 12 V, the LED lighting and sensors operate with 5 V and logical CMOS level of 3.3 V. The user interface was designed in order to be intuitive and waterproof with no leaking fissures to the interior of the device. Thus, the capacitive interface was selected with a dedicated microcontroller STM32F091CBT6 (IC2) ARM^®®^ 32-bit Cortex^®®^ M0 CPU frequency up to 48 MHz for both RGB led lighting (D1~D8) and capacitive detection. The main control of the operation was performed by STM32F303RET6 (IC1) ARM^®®^ Cortex^®®^ M4 32-bit CPU with 72 MHz FPU, which controlled the overall firmware architecture. Therefore, IC1 was able to control the operation of the motor with the aid of an H-Bridge DRV8872-Q1 (IC3) with a high range of operation up to 3.6 A and 45 V. In addition, it made the interface communication with IC2 and Bluetooth HM−10 (IC7) through UART, LCD screen design through SPI, sensors input through ADC channels and system-timings for culture operation. In order to measure the temperature, an LMT85 sensor (IC4) and magnetic field (AD22151) (IC5) in the most adequate position an extra PCB was designed in order to house both sensors and wire communicate with the main board for a closed-loop system response. A linear position sensor 9610R3.4KL2.0 (IC6) was used in order to access magnets platform position to control permanent magnets positioning and resulting magnetic field, was easily integrated with an ADC channel input. All integrated circuits were designed with their respective decoupling capacitors in order to avoid high-frequency power transitions to dwell in the rest of the circuit power lines. The 5 V regulator LM2596 (IC9) was a switching step-down with 80% efficiency, being important for this application by comparison with linear regulators, during long periods of use the amount of heat produced is considerably less for this level of power required. In order to power the logic circuits at 3.3 V, an LD1117 voltage regulator (IC10) was employed.

### 2.2. Firmware and Interface Design

The developed circuit was based on microcontrollers and digital communication with every component on the control boards. Regarding magnetic output, it must be noted that the peak amplitude was topped by the permanent magnets magnetization grade, thus the electromechanical component will control solely how much amplitude will reach the culture plate through the Hall sensor, as well as the displacement and frequency. Thus, the resulting mechanism control followed an approach by calculated displacement steps that will fit a distance at a given displacement frequency in order to output at the culture plate an approximate alternated magnetic field wave. However, motor sliding was a problem, using firmware breaking mechanisms with the aid of the IC3 H-Bridge, together with linear sensor for displacement error, more precise control was enabled over the mechanical response, and consequently the magnetic field. The interface, communication and each machine state (Figure 5) was controlled by firmware developed for each microprocessor according to their own respective tasks. The different machine states allowed the user to interface with each system functionality and take advantage of its calibration, variable settings and running processes.

The device works through four main states: (i) a menu state where every state returns to and consists of the core control of the device; (ii) a calibration state where the user sets the starting position; (iii) a program calibration state where the user sets the culture variables; (iv) a running state where the machine performs the programmed culture by the user. Through the capacitive interface, the developed system allows us to navigate a menu to adjust variables (Table 1) such as whole culture duration; active and resting cycles duration which work through two temporal levels (short and long cycles); table position calibration and setting starting point. It is possible to store up to three culture programs through emulated electric erasable programmable read-only memory (EEPROM) in the program memory. Mechanical characteristics of the developed system allows us to move the magnets up to 25 mm at a speed of up to 2.5 mm per second. The firmware provides these limitations in order to protect the cells and hardware from human error by calculating hardware limits according to distance and operation frequency. The user can define various stimulation parameters such as critical temperature, displacement between 1 and 25 mm, resulting in operating frequencies between 0.1 and 2 Hz and different stimuli cycle timings to adapt the culture to the cell’s native environments.

## 3. Bioreactor Evaluation

The performance of the herein developed bioreactor was evaluated using magneto-active materials based on composites comprising the piezoelectric poly(vinylidene fluoride) (PVDF) and the magnetostrictive Terfenol-D (TD) particles (Etrema Products) with approximately 1 µm diameter, as described in References [20,45]. This material was selected due to its magnetoelectrical properties, i.e., actively responding to the magnetic field provided by the magnetic bioreactor. Due to their magnetostrictive component (TD), the material senses the magnetic field, inducing a mechanical stimulation on PVDF, which due to its piezoelectric properties further induce an electrical polarization variation, creating the electrically active microenvironment that is translated to the cells [46].

MC3T3-E1 pre-osteoblast cells (Riken Bank) were used for the cell proliferation assays, as a proof of concept for bone regeneration studies. Previous to the cell culture studies, the cells were maintained in Dulbecco’s modified Eagle’s medium (DMEM from Gibco, ThermoFisher, Loughborough, UK) containing 1 g L^−1^ glucose, 10% fetal bovine serum (FBS from Biochrom, Cambridge, UK), and 1% penicillin/streptomycin (P/S, Biochrom) in a controlled atmosphere at 37 °C and 5% CO_2_. The culture medium was replaced every 2 days, and at pre-confluence, cells were harvested using trypsin−ethylenediaminetetraacetic acid (EDTA)(Biochrom). Non-poled (non-charged) films were used to study the effect of the mechanical stimuli provided by the magnetostrictive particles in cell proliferation while poled films were used to study the influence of the mechano-electrical stimulus provided by the combination of electroactive PVDF and TD particles. ME films with a diameter of 1.3 cm were sterilized using UV for 1 h each side and placed in a 24-well tissue culture polystyrene plate. Then, 3 × 10^4^ cells∙mL^−1^ in DMEM were seeded on each well and incubated for 24 h. For this, a drop method was used, in which approximately thirty-five microliter of DMEM containing 15,000 cells was first placed on the surface of the material for 30 min to allow the cell adhesion, and then 250 μL DMEM was added to the well. After 24 h incubation time, one plate was used for the static cell culture (without any stimulation) and the other was transferred onto the bioreactor for 48 h at 37 °C in a 95% humidified air containing 5% CO_2_, totalizing two cycles of magnetic stimulation. The dynamic stimuli provided by the magnetic bioreactor was achieved through the following procedure: an active time of 16 h under magnetic stimuli, which was divided into intervals of 5 min active stimuli and 25 min of resting followed by a period of complete inactivity of 8 h (Figure 6a). After 48 h, the 3-(4,5-dimethylthiazol-2-yl)-5-(3-carboxymethoxyphenyl)-2-(4-sulfophenyl)-2 H-tetrazolium (MTS, Promega) assay was used in order to determine the cell viability at the defined time-points. MTS assay is a coloring method that allows determining the cell viability and is based on the NADPH or NADP-assisted bioreduction in living cells. For this assay, the samples were transferred to a new 48-well plate and further incubated with an MTS solution (in a 1:5 ratio) at 37 °C and 5% CO_2_. After 2 h, 100 μL of each well was transferred to a 96-well plate, and the optical density (OD) of each well was measured at 490 nm using a spectrophotometric plate reader (Synergy HT from BioTek, Colmar Cedex, France).

The selected conditions were employed in order to resemble the human body’s daily mechanical conditions divided by 16 h of activity and 8 h of sleep, also considering the fact that bone is piezoelectric itself [21] and the magnetoelectrical scaffold is able to mimic the electroactive microenvironment upon magnetic stimulation. Those short bursts of stimuli for a duration of 5 min were performed by displacing 25 mm the magnetic table at a frequency of 0.3 Hz, resulting in a magnetic field variation of up to 23 mT within the cell culture wells. For every studied condition, three samples were assayed, and growing cell viability was determined through the MTS assay. For this assay, three main variables were considered: (i) under static conditions, i.e., without magnetic stimulus and bearing in mind the single effect of the different morphologies related to pore size differences, (ii) under dynamic conditions considering magnetic stimuli effect, and (iii) the relative effect between the material surface charge and magnetic stimuli.

The bioreactor system was found to be completely biocompatible and suitable for cell culture. After 48 h of cell culture, MTT results show that cells grow and proliferate independently of the condition applied, showing more than 100% of cell growth under all conditions. All the components are thus biocompatible and the system working properly to avoid the increase in the temperature, one of the main concerns related to this device.

Different conditions applied to the scaffolds further induce different effects in terms of cell proliferation. The application of magnetic stimuli brings an unequivocal increase of proliferation rate in all samples, indicating a clear response of the ME films to the magnetic field, thus demonstrating that the bioreactor provides a suitable microenvironment to the pre-osteoblast cells especially in positively charge TD/PVDF film (Figure 6b). The clear increase in cell viability upon application of the stimuli indicates that a mechanoelectrical effect occurs on non-poled samples while a magnetoelectric effect occurs on poled samples, being the later more beneficial for cell growth. On all tested magneto-responsive materials, statistically significant differences in proliferation rate were observed on the growing cells.

## 4. Conclusions

There is an ever-increasing need for more efficient strategies in TE applications. This fact is becoming a driving force in the R&D efforts to develop a new class of materials, smart materials that respond to stimuli that further triggers appropriate cellular response through the creation of a proper microenvironment. The complexity of these microenvironments where cells are able to optimize cell growth and control cellular functions, make it difficult to recapitulate in vitro. Thereby, the need for these materials and devices capable of recreating in vivo conditions are key elements for the next developments in TE applications.

The magnetic bioreactor developed in this study represents an advance on the state of the art in TE and provides, together with specific magnetoelectric scaffolds, the electrically active microenvironments necessary for cell tissue regeneration. The bioreactor was designed and constructed taking into consideration the envisaged operating principles, by using biocompatible materials, conventional mechanics and digital electronics. It also has the potential to integrate other electronic modules that support digital communication for synchronization. This study further proved that tissue cultures may be performed with this system since a boosting effect on the proliferation rate was observed upon application of the stimuli and no signs of toxicity were found. Simultaneously, this experiment demonstrates the suitability of magneto-responsive scaffolds for adhesion and proliferation of pre-osteoblasts, availing itself from the mechanical and electrical microenvironment conceived in the material. It was possible to conclude that the magnetic module of this bioreactor was able to provide an important contribution to building the proper microenvironment as a device, whereas a scaffold which provides the proper cues to cells with the physical environment, mechanical and electrical stimuli that can be used synergistically with the system.

Therefore, this work has provided the development of a novel bioreactor based on magnetic stimulation that has proven that the developed bioreactor is biocompatible and that may be used for advanced tissue engineering applications, allowing for advanced tissue engineering strategies. It will certainly act as a valuable tool for mimicking in vitro the human stimulations provided by the electrically active tissues that are present in the body. It could also be important for growing well-formed cellular tissues in vitro in a more effective and rapid way, which could be further implanted in the human body without the material. In the case that magnetoactive materials to be implanted in the human body, it would provide a suitable platform to evaluate the remote stimulation and thus for effective growth and differentiation of cells in immobilized patients. In fact, mimicking cell microenvironments is thus a key issue to recapitulate in vitro what occurs in vivo and this bioreactor holds great promise to fulfill such requirements.

## Figures and Tables

**Figure 1 sensors-20-03340-f001:**
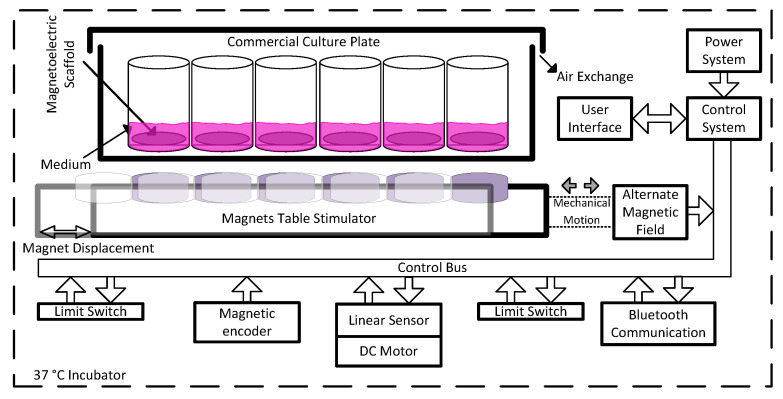
Magnetoelectric bioreactor operating principle through the use of electrical and mechanical controls to produce an alternated magnetic field and thus stimulate the magnetoelectric scaffolds and, consequently, the cells.

**Figure 2 sensors-20-03340-f002:**
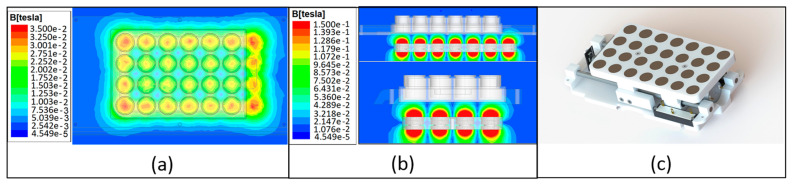
(**a**) Magnetic field intensity distribution at the bottom of 24-wells cell culture plates, (**b**) magnetic field force lines simulation in frontal and side planes and (**c**) rendered model of the mechanical permanent magnets table ball-screw assembly.

**Figure 3 sensors-20-03340-f003:**
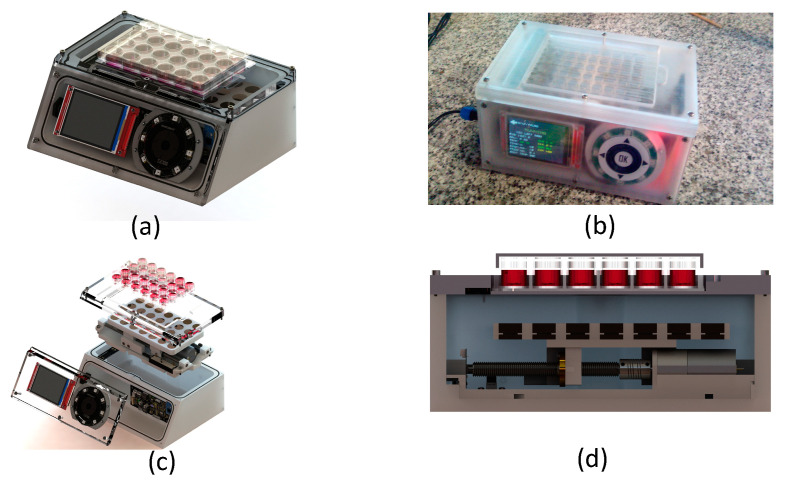
(**a**) Representation of the bioreactor assembled with a cell culture plate; (**b**) Bioreactor prototype built mechanism with every component; and schematic representation of (**c**) all disassembled main electric and mechanical components and (**d**) of the mechanical component represented as a transversal cut.

**Figure 4 sensors-20-03340-f004:**
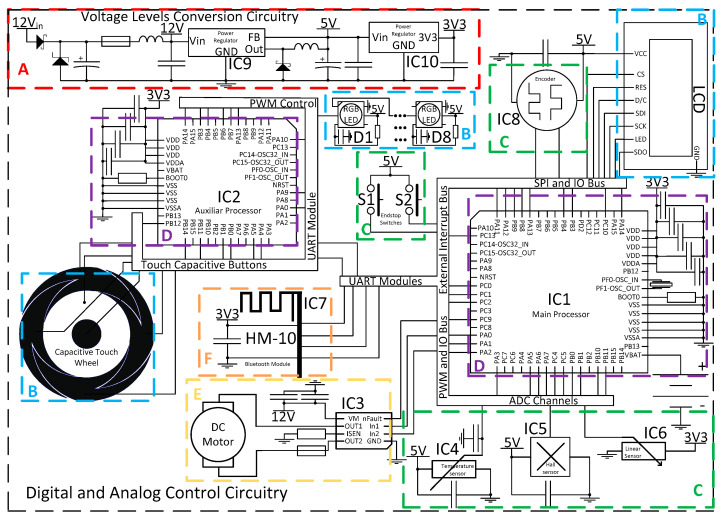
Main circuits used for the power conversion (**A**), user interface (**B**), sensors (**C**), system control (**D**), actuators (**E**) and wireless communications (**F**).

**Figure 5 sensors-20-03340-f005:**
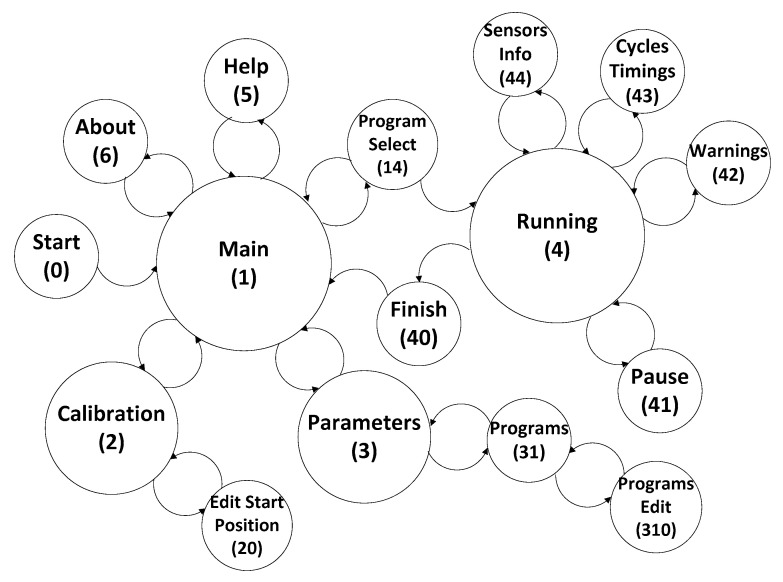
State machine control nodes.

**Figure 6 sensors-20-03340-f006:**
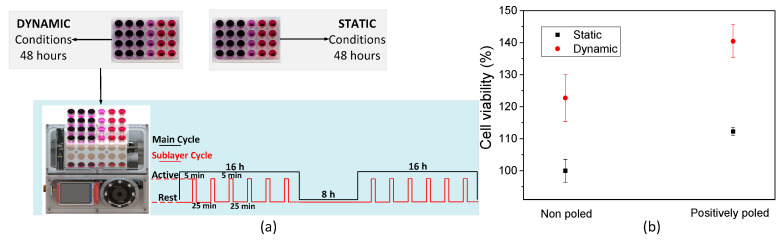
(**a**) Stimuli schedule timing programmed in the bioreactor for pre-osteoblast tissue culture assays using either static or dynamic conditions and (**b**) cell viability after 48 h of cell culture on TD/PVDF films with and without magnetic stimuli. The cell viability was calculated regarding the cells growing on the non-poled ME film at static conditions presented as % of growth. In each study, three samples were assayed per studied condition.

**Table 1 sensors-20-03340-t001:** Bioreactor user control variables and respective ranges to be set in the programs menu.

User Control Variables	Description	Ranges
Displacement	Distance traveled by the respective permanent magnet platforms.	5–25 mm
Frequency	Frequency of stretch or magnetic field stimuli signal to be applied.	0.1–2 Hz
Runtime	Culture total running time.	1–180 d
Cycle 1 active time	Active time of sublayer cycle included in the main layer active time.	1–360 min
Cycle 1 resting time	Resting time of sublayer cycle included in the main layer active time.	1–360 min
Cycle 2 active time	Active time of the main layer cycle.	1–24 h
Cycle 2 resting time	Resting time of the main layer cycle.	1–24 h
Shutdown temperature	Temperature value, which shuts down stimuli until culture temperature lowers to safety values again.	30–40 (°C)

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
