# Peer review of "Magnetic Bioreactor for Magneto-, Mechano- and Electroactive Tissue Engineering Strategies"

_sensors, 2020, doi:10.3390/s20123340_

Round 1

Reviewer 1 Report

In this manuscript a novel bioreactor system for bone tissue engineering is described. The novelity of this system is the combination with electromechanical stimuli for cell-seeded scaffolds that can be pre-incubated before implantation into the patient. Further, the authors mention that this system could be applied in the clinical practice in immobilizied patients since it allows a remote stimulation of cells without direct contact. Overall, it is well written manuscript. Castro et al. provide a good and informative introduction into this topic and describe their bioreactor system in great detail. Provided figures help to understand this system. As proof-of-principle a small in vitro cell culture study was performed. This could have been done in greater detail, e.g. using different kind of cells / longer incubation periods / thicker 3D scaffolds. However, in my mind this system itself is extremely interesting for the readers of Sensors and publishing at this point can be recommended. However, I would suggest that the authors give some more details about the possible application in clinical practice since this is stated in the abstract, but was not mentioned anymore in the manuscript. Further, it would be interesting to get information how long scaffolds (perhaps depending on the size?) should be incubated in this system before implantation in vivo.

Reviewer 2 Report

The authors presented a novel bioreactor that allow to magnetically stimulate magnetoelectric scaffolds and claim to provide mechanical and electrical stimuli to the cells though magnetomechanical or magnetoelectrical effects. Some details on cell culture part would need to be properly considered:

  • detail on cells maintenance and the culture component, i.e. medium, how many days the cells were cultured and fed before seeding step need to be included. 
  • the author presented a cell viability graph, the Y scale on the graph showing cell viability in percentage? The graph need to be properly labelled. 
  • dynamic and static phrase normally referring to feeding strategy of the cells in bioreactor; dynamic culture happen in perfusion condition where there is a flow change of culture medium, static culture refer to no flow change of the medium, whilst in this study, the author used those terms to refer to magnetic stimulation during the culture, which is a bit confusing. Perhaps could consider to change it to i) stimulated and ii) non-stimulated groups since both groups were cultured in static condition. 
  • the magnetic stimulation step on the cells was done for 5 minutes, resting gap for 25 minutes, total time for 1 cycle to completed was 24 hours before continue to next stimulation. Does MTS assay done at 48 hours were counted from 24 hours seeding cells on scaffold + 24 hours after 1 cycle stimulation completed? Or after 48 hours stimulation cycle? Also only one time point was assessed at 48 hrs, which is less conclusive, at least should have 2-3 time points to compare, i.e. 48,72, etc..Also does the rate used for magnetic stimulation is constant for every repeat? 
  • In MTS assay, the author mentioned on static conditions, i.e. without magnetic stimulus and bearing in mind the single effect of the different morphologies related to pore size differences.. Any microscopy analysis/image  showing the morphological difference on cells? 
  • The author mentioned different conditions applied to the scaffolds further induce different effects in terms of cell proliferation. Does the stimulation condition applied to the scaffold before seeding the cells?, or together with the cells? Would it be any difference on the cells growth if the stimulation on scaffold was done prior to the seeding process? In case the stimulation give direct effect on the cells, or the scaffold itself..Perhaps another tested group need to be considered here. 

Reviewer 3 Report

Magnetic Bioreactor for Magneto-, Mechano- and Electroactive Tissue Engineering Strategies

In this work, a bioreactor has been designed and built allowing to magnetically stimulate magnetoelectric scaffolds and therefore provide mechanical and electrical stimuli to the cells though magnetomechanical or magnetoelectrical effects, depending on the piezoelectric nature of the scaffold. Proposed magnetic bioreactors allows for a remote stimulation without direct contact with the material. This could be valuable to mimic what occurs in clinical practice and for application in immobilized patients.

I found that the manuscript is interesting and have some merits. In my point of view, the manuscript could be considered for acceptance but not in its current form. Having said that following revisions are suggested; It includes some scientific as well as presentation issues, some errors and I list the comments to improvise the manuscript below:

Comments:

  1. The abstract is descriptive and qualitative. Normally an abstract should state briefly the purpose of the study undertaken and meaningful conclusions based on the obtained results. Hence, this needs rewriting. I would expect brief, yet concise, the quantitative data description of the results in the abstract.
  2. The given list of keywords is superficial with broader terms. More specific terms should be used. Replace accordingly.
  3. The introduction is adequately expressed. However, some more recent and relevant literature can be added.
  4. The English used is not up to the standard of the journal. Some sentences are long and badly worded with repetitive words. Please consider breaking longer sentences into smaller fragments for easy understanding.
  5. The novelty of the study should be clearly highlighted in the manuscript at the end of the introduction section, as there are some existing literature reports.
  6. Some Figures are poorly constructed as Figure 5 is blurry and difficult to read. Reconstruction is required.
  7. The conclusion is superficial. Herein, I would like to see the major findings and how they are addressing the left behind research gaps and covering current challenges.
  8. Referencing is not right. Literature needs to be updated with care. At least 20% references should be from recent years 2018-2020.

Round 2

Reviewer 2 Report

The author mentioned that the present work is just a validation of the bioreactor, not a tissue engineering cell evaluation, it would be good to have data on cell morphology before and after stimulation on the scaffold, as it does involve cell evaluation also referring to the title used on the paper for tissue engineering strategies, unless if the author could make a slight change on the title used.
